# P3CMQA: Single-Model Quality Assessment Using 3DCNN with Profile-Based Features

**DOI:** 10.3390/bioengineering8030040

**Published:** 2021-03-19

**Authors:** Yuma Takei, Takashi Ishida

**Affiliations:** 1Department of Computer Science, School of Computing, Tokyo Institute of Technology, Ookayama, Meguro-ku, Tokyo 152-8550, Japan; takei@cb.cs.titech.ac.jp; 2Real World Big-Data Computation Open Innovation Laboratory (RWBC-OIL), National Institute of Advanced Industrial Science and Technology (AIST), Aomi, Koto-ku, Tokyo 135-0064, Japan

**Keywords:** model quality assessment (MQA), estimation of model accuracy (EMA), protein structure prediction, machine learning, deep learning, 3DCNN, CASP

## Abstract

Model quality assessment (MQA), which selects near-native structures from structure models, is an important process in protein tertiary structure prediction. The three-dimensional convolution neural network (3DCNN) was applied to the task, but the performance was comparable to existing methods because it used only atom-type features as the input. Thus, we added sequence profile-based features, which are also used in other methods, to improve the performance. We developed a single-model MQA method for protein structures based on 3DCNN using sequence profile-based features, namely, P3CMQA. Performance evaluation using a CASP13 dataset showed that profile-based features improved the assessment performance, and the proposed method was better than currently available single-model MQA methods, including the previous 3DCNN-based method. We also implemented a web-interface of the method to make it more user-friendly.

## 1. Introduction

Protein structure prediction is one of the most important issues in bioinformatics, and various prediction methods have been developed such as Alphafold [1]. Currently, there is no best structure prediction method for any target, and thus, the user practically uses multiple methods to generate structure models. In addition, each method often generates multiple structure models. Therefore, model quality assessment (MQA), which estimates the quality of the model structures, is required for selecting a final structure model. There are two kinds of MQA, single-model and consensus method. The single-model methods use only one model structure as the input, whereas the consensus methods need to take a consensus of multiple model structures. The performance of consensus methods is often higher in a case that many high-quality model structures are available, such as CASP [2]. However, consensus methods do not perform well with fewer model structures. Furthermore, the consensus methods use single-model MQA scores for the features [3]. Therefore, the development of high-performance single-model MQA methods is important.

To improve the single-model MQA, various methods have been developed. Currently, one of the best methods is based on the three-dimensional convolutional neural network (3DCNN) [4,5]. 3DCNN is a derivative of deep neural networks and can handle 3D-coordinates of protein atoms directly. However, these methods show only comparable performance to the other existing methods. One of the major problems in the previous 3DCNN-based methods was with their features, which were based on only atom types. They did not use any features related to evolution, such as sequence profile information. Various studies have shown the importance of sequence profile information, and the addition of such information could significantly improve the performance [6,7].

In this study, we developed a single-model MQA method for protein structures based on 3DCNN using sequence profile-based features. The proposed method, profile-based three-dimensional neural network for protein model quality assessment (P3CMQA), has basically the same neural network architecture as that of a previous 3DCNN-based method [5] but uses additional features, i.e., sequence profile information, and predicted local structures from the sequence profiles. Comparison with state-of-the-art methods showed that P3CMQA performed considerably better than them. P3CMQA is available both as a web application and a stand-alone application at http://www.cb.cs.titech.ac.jp/p3cmqa and https://github.com/yutake27/P3CMQA, respectively.

## 2. Materials and Methods

A 3DCNN-based method by Sato and Ishida (Sato-3DCNN) used only 14 atom types as the features [5]. In addition to the atom-type features, here, we tried to improve the performance by adding sequence profile-based features. We used a position-specific scoring matrix (PSSM) of a residue obtained by PSI-BLAST [8] to incorporate evolutionary information. We also added predicted local structures from the sequence profiles: predicted secondary structure (predicted SS) and predicted relative solvent accessibility (predicted RSA). SS and RSA are predicted by SSpro/ACCpro [9]. The overall workflow is shown in Figure 1.

### 2.1. Featurization

#### 2.1.1. Making Residue-Level Bounding Box

We made a residue-level bounding box in the same way as in Sato-3DCNN. The length of one side of the bounding box is 28Å and the box is divided into 1Å voxels. The axis of the bounding box is defined by the orthogonal basis calculated from the C-CA vector and N-CA vector and the cross product of C-CA and N-CA. By fixing the axis, the problem of rotation need not be considered.

#### 2.1.2. Atom-Type Features

In Sato-3DCNN, 14 features corresponding to combinations of atoms and residues based on Derevyanko-3DCNN [4] were used. The detail of the 14 atom-type features is shown in Table A1. In this work, we used all 14 features as atom-type features and all these features are binary indicators for each voxel.

#### 2.1.3. Evolutionary Information

We used the position-specific scoring matrix (PSSM) as evolutionary information. We generated PSSM using PSI-BLAST [8] against the Uniref90 database (downloaded April 2019) with two iterations. The maximum and minimum values of PSSM in the training dataset were 13, −13; thus, we normalized PSSM using the following formula.
$$Normalized\;PSSM=\frac{PSSM+13}{26}$$

PSSM is a residue-level feature, but we assigned PSSM to all the atoms that made up the residue.

#### 2.1.4. Predicted Local Structure

We used predicted local structure as a feature, and the actual local structure of the model structure was not used because it is considered to be observable using 3DCNN. Predicted secondary structure (SS) and predicted relative solvent accessibility (RSA) were used as the predicted local structure. SS was predicted from the sequence profile using SSpro [9]. SSpro predicts SS into three classes; therefore, we use predicted SS in the form of a three-dimensional one-hot vector. RSA was predicted from the sequence profile using ACCpro20 [9]. ACCpro20 predicts RSA from 5% to 95%, as a 5% increase; thus, we divided the predicted RSA by 100 and scaled it from 0 to 0.95. Like PSSM, predicted SS and RSA are residue-level features, but we assigned them to all atoms.

### 2.2. 3DCNN Training

#### 2.2.1. Network Architecture

The same network architecture used in Sato-3DCNN, which consists of six convolutional layers and three fully connected layers, was used for the neural network architecture. To avoid overfitting, the batch normalization [10] was applied after each convolutional layer, and PReLU [11] was applied as an activation function after the batch normalization. We show the detail of the network architecture in Table A2. We examined other network architectures such as residual network [12], but they did not improve performance significantly.

#### 2.2.2. Label and Score Integration

We trained our models by supervised learning. We generated a bounding box for each residue and trained the model; hence, a label is required, which represents the local structure quality of each residue. Thus, we used lDDT [13] as a label for each residue, as with Sato-3DCNN. lDDT is a local structural similarity score that is superposition-free. It evaluates whether the local atomic environment of the reference structure is preserved in a protein model.

We trained models by binary classification, as in Sato-3DCNN. Therefore, we set the threshold to 0.5 and converted lDDT to positive or negative. Sigmoid cross entropy was used as a loss function. We also tried regression learning, but the predictive performance decreased.

As mentioned above, we trained a 3DCNN-model for each residue, and the 3DCNN-model returned a score for a residue in the prediction. Thus, we needed to integrate the predicted score of each residue into the score of the entire protein structure model to predict its quality. The score of the entire protein structure model was calculated as the mean of the score for each residue, as with Sato-3DCNN.

In this study, we compared our method to other methods to predict the quality of the entire protein structure model. We used GDT_TS [14] as a label that represents the quality of the entire structure model in the evaluation.

#### 2.2.3. Parameters

We used AMSGrad [15] as an optimizer and set the learning rate to 0.001. We used 32 Nvidia Tesla P100 GPUs and performed distributed learning. We set the batch size of each GPU to 32, and the total batch size was 1024.

#### 2.2.4. Training Process

We trained the model using the architecture, labels, loss function, and optimization function described above. To avoid overfitting, the average Pearson correlation for each target of the validation dataset was calculated for each epoch, and the model of the epoch with the best Pearson correlation was selected. The training of the model took about two hours for each epoch and was completed within 5 epochs.

### 2.3. Dataset

For the training dataset, we used the predicted protein model structures and native structures from CASP7 to CASP10 [16,17,18,19]. The total number of protein targets was 434, and the total number of protein structure models was 116,227. We excluded the protein targets with fewer than 50 models. Finally, the training dataset included 421 protein targets and 116,096 protein structure models. We randomly split each protein target into the training dataset and the validation dataset (8 to 2). In the training dataset, due to a large number of structure models, we randomly selected 25% models for each protein target. As a result, we used 23,405 structure models and 4,666,496 residues for training.

For the test dataset, we used CASP12 and CASP13 datasets [2,20]. Each dataset was divided into stage 1 and stage 2. However, only stage 2, which has 150 structure models per protein target, was used. The GDT_TS for each structure model was obtained from the CASP website. In CASP, protein targets for which the best-predicted model had GDT_TS less than 40 were not considered in the evaluation. Thus, we excluded such protein targets, and finally, the number of protein targets in CASP12 and CASP13 was 51 and 66, respectively.

In both the training dataset and the test dataset, we used SCWRL4 [21] to optimize the side-chain conformation. By using SCWRL4, it is possible to evaluate the quality of model structures that contain only main-chain. The details of both datasets are shown in Table 1.

### 2.4. Performance Evaluation

We used the following four measures to evaluate the performance of our method.

The average Pearson correlation coefficient for each targetThe average Spearman correlation coefficient for each targetThe average GDT_TS loss for each targetThe average Z-score for each target

Pearson correlation coefficient is calculated from the correlation between the predicted quality score and the GDT_TS. Spearman correlation coefficient is calculated from the same data as the Pearson correlation coefficient. GDT_TS loss is the difference between the GDT_TS of the best model and the GDT_TS of the model with the highest prediction score. Therefore, a lower GDT_TS loss represents better performance. Z-score is a standard score of a selected model, which is the difference between the GDT_TS of a selected model and the population mean in units of standard deviation. A higher Z-score indicates a better performance.

## 3. Results and Discussion

### 3.1. Training Result for Each Feature

We added evolutionary information and predicted local structural features to improve the performance of the model. Thus, we evaluated the contribution of each feature using the validation dataset. To compare the prediction performance of trained 3DCNN-models using each feature, the average of the Pearson correlation between the predicted global score and GDT_TS for each target was used because we selected the model with the best Pearson correlation on the validation dataset during training. We show the results in Table 2. The model using only atom-type features was the same as Sato-3DCNN except for the split between the training and validation sets and the optimizer. This result shows that the prediction performance was greatly improved by adding profile-based features. Both evolutionary information and predicted local structural features contributed to improving the performance, and the latter had a slightly larger contribution. When we used all these features, the best performance was achieved. The results for metrics in addition to the Pearson correlation on the validation dataset are shown in Table 3. For the other metrics, the best performance was obtained when all features were used.

### 3.2. Comparison with Other Methods on CASP Datasets

We compared the proposed method with major single-model MQA methods on the CASP12, CASP13 datasets. We used Sato-3DCNN [5], ProQ3D [22], SBROD [23], and VoroMQA [24] as comparison methods. Sato-3DCNN is the direct predecessor of this research. ProQ3D is a deep neural network-based method using profile-based features. SBROD is a method using ridge regression with various geometric structural features. VoroMQA is a method using a statistical potential that depends on interatomic contacts by Voronoi tessellation.

The results of all comparison methods were executed by us. Sato-3DCNN is a retrained model using the same training data as that of this method with an optimizer changed from SMORMS3 [25] to AMSGrad. ProQ3D, which was last updated on 8 October 2017, was downloaded, and we used the S-score version model. The version of VoroMQA was 1.19.2352. SBROD, which was last updated on 14 August 2019, was downloaded, and we used the model that was trained using CASP5-10 for the training dataset.

The result for the CASP12 stage 2 dataset is shown in Table 3. The proposed method showed better performances for each metric. To check whether there is a statistically significant difference between the performance of the proposed method and the comparison method, we conducted the Wilcoxon signed-rank test at a significance level of 0.01. As a result, it is shown that there are statistically significant differences between the proposed method and the comparison methods in terms of Pearson and Spearman correlation. For GDT_TS loss and Z-score, there is no significant difference, but the proposed method performed better than the other methods. Also, the results for the CASP13 stage 2 dataset is shown in Table 4. Similarly, the results for CASP13 stage 2 showed that the proposed method was better than the other existing methods.

We also compared the performance for each category of targets that represent the difficulty of the prediction released by CASP. We used three categories: Free Modeling (FM), Free Modeling/Template-Based Modeling (FM/TBM), and Templated Based Modeling (TBM). The average Pearson correlation coefficient for each category in CASP13 are shown in Table A4 and Figure A1. The proposed method is the best for each category. For FM/TBM and TBM categories, there are significant differences between the proposed method and the other methods. For the FM category, there is no significant difference due to the small number of targets (12 targets).

## 4. Web Tool

The proposed method showed the best performance as a single-model MQA method. However, it is impractical if it is not readily available to the user. For example, ProQ3D [22] provides a web interface, but it simply outputs the global score for the entire model structure and the local score for each residue as text. Thus, we implemented a web-based tool to make it even more user-friendly to check the results.

Figure 2 shows the input page of the web tool. The required inputs are an email address and a model structure in PDB format or mmCIF format. A target sequence in FASTA format can be entered as an option. If the sequence is not entered, a sequence generated from the model structure file is used to construct the profile-based features. When the prediction is finished, users will receive an email with an URL of the prediction result.

The execution time depends on the size of a protein. For a new protein, it takes a little longer because it is necessary to generate a sequence profile. If the sequence length is about 500, it will take about 30 min to complete. However, for proteins that have been processed before, the profile generation can be skipped, and the execution time is about one minute.

Figure 3 shows an example of the output of the web tool. It outputs a global score and a bar chart of residue-wise local scores. A local score is an estimation of lDDT and it takes values between 0 and 1. A higher value shows a better model structure. It also provides a 3D view of a model structure colored according to the prediction score for each residue. The model structure is colored in rainbow colors, with red areas representing low local scores and blue areas representing high local scores. It uses NGL viewer [26], and thus, the user can move and rotate the model structure to visually check the quality.

Besides, the predictions can be downloaded in several formats. It is possible to download a PDB file with prediction scores set to the b-factor of each residue. This makes it possible to check the detailed structure and prediction scores in your local environment.

## 5. Conclusions

We developed a single-model MQA method based on 3DCNN, called P3CMQA. It used sequence profile-based features and performed better quality estimation than that of existing single-model MQA methods. We have also developed a web tool of the proposed method, which provides prediction results in a user-friendly format.

## Figures and Tables

**Figure 1 bioengineering-08-00040-f001:**
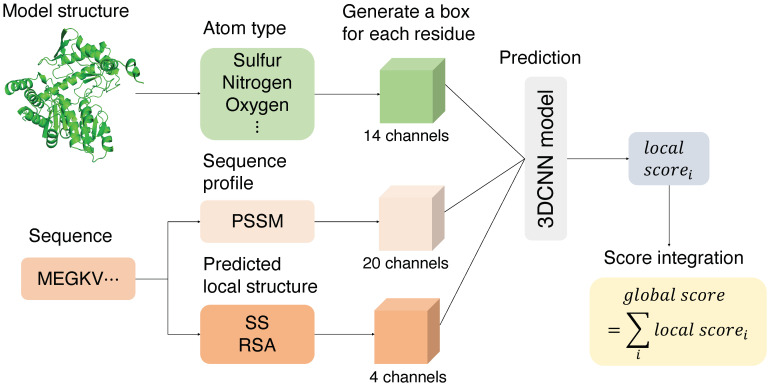
Overall workflow of this work. First, a bounding box is generated for each residue from the coordinate information of the model structure. Then, 14-dimensional atom-type features are obtained from the model structure. In addition, 20-dimensional sequence profile features and 4-dimensional local structure features are generated from the sequences. These features are then input to the three-dimensional convolutional neural network to predict a local score for each residue. Finally, the local scores are averaged to obtain a global score for the entire model.

**Figure 2 bioengineering-08-00040-f002:**
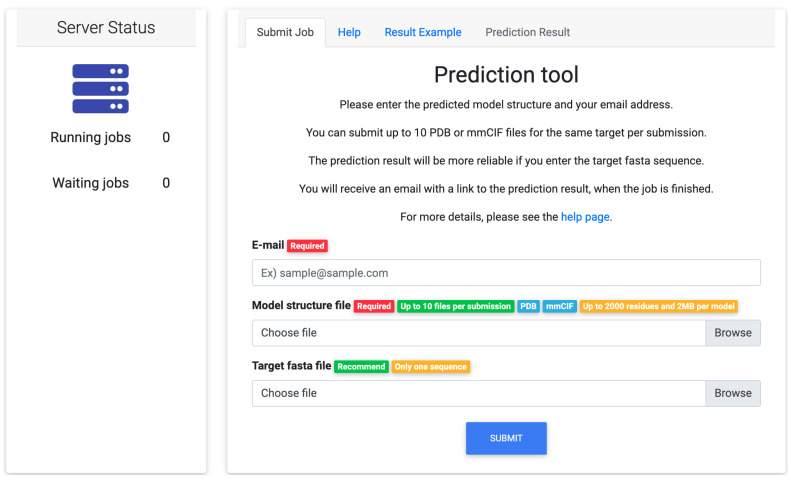
The input page of the web tool. The email address and the model structure in PDB format or mmCIF format are required inputs, and the sequence in FASTA format is an optional input. You can check the number of running jobs and the number of waiting jobs.

**Figure 3 bioengineering-08-00040-f003:**
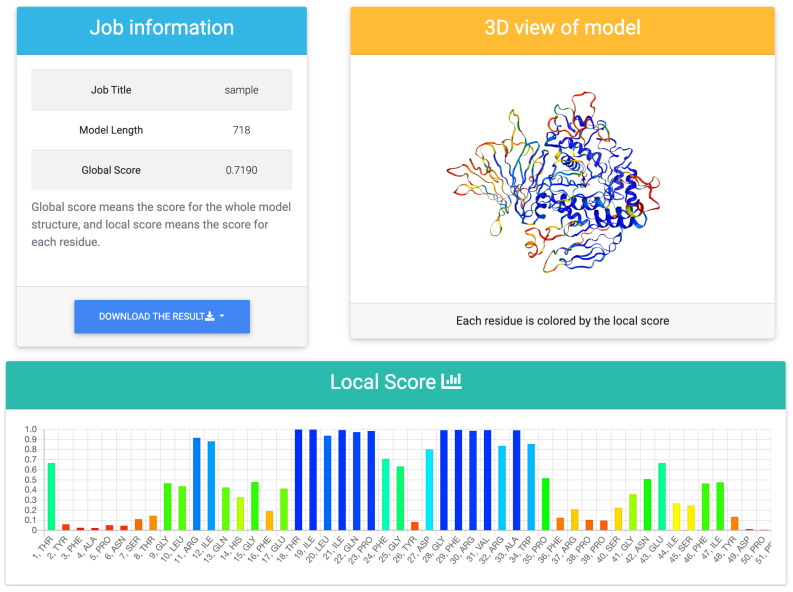
The output page of the prediction results. The predicted score for the whole model, the predicted score for each residue, and the three-dimensional structure colored by the local score is shown. The parts colored in blue represent high local scores, and the parts colored in red represent low local scores. The results can be downloaded in multiple formats.

**Table 1 bioengineering-08-00040-t001:** Details of the training dataset and the test dataset.

	Dataset	Number of Targets	Number of Model Structures per Target
Train	Train	337	69.5
	Validation	85	271.7
Test	CASP12	51	149.9
	CASP13	66	149.9

**Table 2 bioengineering-08-00040-t002:** Prediction performance for the combination of each feature.

Atom-Type Features	Evolutionary Information	Predicted Local Structure	Pearson (Validation)
✓	✗	✗	0.757
✓	✓	✗	0.834
✓	✗	✓	0.847
✗	✓	✓	0.858
✓	✓	✓	0.865

Columns 1–3 represent combinations of features referenced in Section 2.1. The fourth column represents the average Pearson correlation coefficient for each target in the validation dataset. The best performance in the fourth column is in boldface.

**Table 3 bioengineering-08-00040-t003:** Performance in the CASP12 stage 2 test dataset.

Method	Pearson	Spearman	Loss	Z-Score
**Proposed**	0.856	0.782	4.319	1.240
	(−)	(−)	(−)	(−)
Sato-3DCNN (AMSGrad)	0.746	0.675	5.530	1.139
	($4.67\times 10^{-9}$)	($5.31\times 10^{-7}$)	($4.89\times 10^{-1}$)	($4.99\times 10^{-1}$)
ProQ3D	0.750	0.672	7.989	0.922
	($8.18\times 10^{-9}$)	($3.41\times 10^{-7}$)	($4.82\times 10^{-3}$)	($7.38\times 10^{-3}$)
SBROD	0.682	0.612	7.063	0.967
	($9.87\times 10^{-10}$)	($1.87\times 10^{-7}$)	($3.47\times 10^{-2}$)	($4.23\times 10^{-2}$)
VoroMQA	0.671	0.592	7.649	0.963
	($1.11\times 10^{-9}$)	($1.77\times 10^{-9}$)	($4.30\times 10^{-2}$)	($4.30\times 10^{-2}$)

The first column represents the method name. The second and third columns show the average Pearson and Spearman correlation coefficients per target. The fourth and fifth columns show the average GDT_TS loss and average Z-score of selected models for each target. The values in parentheses are the *p*-values calculated by the Wilcoxon signed-rank test for the difference between the proposed method and the comparison method. The best values and *p*-values smaller than 0.01 are shown in bold.

**Table 4 bioengineering-08-00040-t004:** Performance in the CASP13 stage 2 test dataset.

Method	Pearson	Spearman	Loss	Z-Score
**Proposed**	0.797	0.757	5.708	1.264
	(−)	(−)	(−)	(−)
Sato-3DCNN (AMSGrad)	0.748	0.703	6.527	1.167
	($1.09\times 10^{-5}$)	($1.84\times 10^{-5}$)	($4.44\times 10^{-1}$)	($3.93\times 10^{-1}$)
ProQ3D	0.686	0.638	9.482	0.990
	($1.42\times 10^{-9}$)	($2.03\times 10^{-10}$)	($2.16\times 10^{-2}$)	($2.29\times 10^{-2}$)
SBROD	0.674	0.637	10.014	0.930
	($1.95\times 10^{-9}$)	($3.38\times 10^{-9}$)	($2.29\times 10^{-4}$)	($5.99\times 10^{-4}$)
VoroMQA	0.676	0.624	12.105	0.786
	($2.38\times 10^{-9}$)	($2.52\times 10^{-11}$)	($1.73\times 10^{-3}$)	($1.15\times 10^{-3}$)

The legends are the same as those in Table 3.

## Data Availability

The web tool can be used at http://www.cb.cs.titech.ac.jp/p3cmqa (accessed on 29 January 2021). The source code and the dataset are available at https://github.com/yutake27/P3CMQA (accessed on 29 January 2021).

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
