# Peer review of "P3CMQA: Single-Model Quality Assessment Using 3DCNN with Profile-Based Features"

_bioengineering, 2021, doi:10.3390/bioengineering8030040_

Round 1

Reviewer 1 Report

After careful reading of the manuscript by Takei and Ishida, I concluded that the merits of the proposed method are not clearly established and several other problems need to be addressed before it is ready for prime time.

The timing may be unfortunate for the authors, but the latest CASP experiment has shown dramatic progress and therefore needs to be taken into consideration, especially that the new CNN-based single model quality assessment method was trained on rather old CASP experiments, with rather different fractions of successful models and different types of failures. 

The features used by the authors, including PSSMs and sequence-based RSA predictions have been used many times before. Hence, it would seem that the main innovation stems from the particular form of convolutional NNs and the novelty here must be clearly delineated and the improvement over simple NNs as baseline demonstrate by direct comparison.

There are many methods for model quality assessment, of which only several were selected for comparison, including another method by the authors which is used to show progress. However, the previous method did not even use PSSM and thus was bound to have limited accuracy. The authors claim that ' Currently, one of the best methods is based on the three-dimensional convolutional neural 32 network (3DCNN) '; yet they admit that 'However, these methods show only comparable performance to the other existing methods', making the same point. The choice of methods for comparison must be carefully justified, and those method should be representative of various classes of successful MQA approaches.

 All tables with results show rather small changes without any analysis of statistical significance or analysis per protein, per class of targets, level of difficulty, regions that are well predicted etc.

The overall architecture, the number of adjustable parameters, training process are described insufficiently, making the overall assessment of the method, strategies to avoid overfitting and analysis of success and failures impossible at this stage.

Categorical RSA prediction methods have long been replaced/made obsolete by real value RSA predictions. By the way, those and the related torsional angle prediction are now largely built into the actual prediction methods that are used to generate the CASP models and hence train the proposed MQA method. How is that affecting the results from older to newer CASP editions?

Reviewer 2 Report

The authors present an improved model-quality predictor. It is closely related to previous method (Sate-3DCNN), but includes additional features that significantly improve the quality of the classification. The method can also be easily used via a website, which makes it easier to use than the Sate method.

The article is well structured and quite legible.

## Article

1. The methods are very concise, and rightly focus on the novel aspects of P3CMQA. However, a little more information about the atom-type features might bring clarity. To me, it was not immediately apparent that the 14 features were per-voxel binary indicators of various atom types.

2. PSSM is a log-odds ratio, so in principle it is unbounded. How are values outside the [-13,13] range from your training set treated? Is the Normalized PSSM simply truncated to 0 or 1?

3. How well do the predicted local scores correlate with lDDT on the test set? The performance evaluation focusses on global GDT_TS, but local quality is also important.

4. It would be interesting to include at least one of the current top-performing MQA methods in the benchmark. For instance, CAMEO winner QMEANDisCo 3 (Studer 2020); CASP14 best single-model method P3De (Wallner); or strong CASP14 method Rosetta (Baker)

5. The performance of Sato-3DCNN on CASP12 (Table 3) is significantly better than that published in Sato 2019. What accounts for the difference? If it is merely test split differences, it may be appropriate to include error bars on the performance based on different splits.

## Software
6. The website is an essential feature. It looks nice and provides the needed functions, as well as conveniences like the NGL pane. I did encounter a few bugs:

    6a. I got errors when submitting a FASTA file with multimeric structures.

    6b. The local score pane did not appear for me when testing a large multimeric structure (https://swissmodel.expasy.org/repository/uniprot/P29973). However it looked fine on a smaller structure.

    6c. Very minor website issue: the email field has 'autocomplete=off' set, which is a little annoying for submitting multiple files.

7. The software should accept mmCIF files. The PDB format has been deprecated since 2014.

8. The website makes sensible privacy decisions (e.g. 5-day expirations). It would be good to add a privacy policy page spelling these out.

9. An open source license should be declared prominantly on github.

10. I encourage the authors to include their method in the CAMEO Model Quality Estimation competition (https://www.cameo3d.org/quality-estimation/) and CASP 15.

Reviewer 3 Report

Takei and Ishida wrote an article about a new single-model quality assessment for 3D protein structures. They based their approach on 3D convolution neural network (3DCNN), but unlike previous attempts by others, their P3CMQA adds profile-based features. They also evaluated how this improves quality assessment. The P3CMCQ method is available on-line as a user-friendly web server.

The topic of model quality assessment is vital, and the authors did a great job motivating their research in the Introduction. The descriptions in Materials and Methods and Results and Discussion are sufficient in general, but I will note some comments later regarding these parts.

I want to praise that Takei and Ishida created a useful web application and made their source code publicly available. Both resources are well-designed and understandable (great Help page and README). For me, this is a significant advantage because many bioinformatics methods nowadays focus only on the manuscript publication and leave the source code messy and web applications challenging to use.

Still, I have some issues regarding the article, which I would like the authors to address:

  • Section 2.1.2 refers to the article by Derevyanko et al. It would be good to summarize the 14 atom-type features. Users would be able to follow P3CMQA approach without necessarily knowing the Derevyanko-3DCNN architecture. The summary may be in the appendix/supplementary.
  • In 2.2.2, a sentence or two about IDDT would help others' understanding of the idea.
  • In 2.3, the authors mention the use of SCWRL4 without giving any rationale for that. Can the authors prove that this is beneficial? What would happen if they used the 3D model structures as they are, without altering the side chains? Does the web application run SCWRL4 for submitted models?
  • Table 2: why does it show only the Pearson correlation coefficient? I would be more interested in Spearman's rho, but more generally, can the table show all four measures of evaluation (Pearson's, Spearman's, GDT_TS loss and the average Z-score)

I also have a list of minor issues to mention:

  • Lines 26-27: "with less model structure" should probably be "with fewer model structures."
  • Should section 2.2.3 be named "Parameters" (plural) or "Parameterization"?
  • Lines 131-132: "Spearman correlation coefficient is calculated in the same way (...)". No, it is not. What the authors probably meant is that it is calculated from the same data.
  • Lines 173: "with the graphics" sound a bit strange and could be rephrased to communicate better.

Round 2

Reviewer 1 Report

While the authors made some efforts to improve the manuscript, including reporting now p-values to assess relative performance of the new method, these changes are largely cosmetic and did not include attempts to improve the study itself and the resulting method.

For example, describing the methods selected for comparison is obviously a marginal improvement over the previous version, but it is not the same as justifying why those methods are representative and sufficient to reach the conclusion of improved accuracy over published works. No attempt was made to include other methods that would likely shed light on the merits and limitations of the proposed approach.

The manuscript still does not define the number of parameters to be optimized and assessment of the training vs. validation vs. test sets, so a couple of generic sentences regarding the philosophy of controlling for overfitting is not sufficient.

The latter point becomes even more critical given that the new method was optimized using the correlation coefficient as the metric of performance, while other methods likely were not (again no comment about that). As can be seen from the modified table with relative performance the new method only achieves statistically significant improvements for the correlation coefficients but not for other important metrics. By the way, there is no discussion of performance within different categories of targets, protein to protein variability, distributions etc. which makes this even more problematic.

In summary, I regard the efforts by the authors as insufficient and missing on the opportunity to make their contribution sound and convincing. Therefore, I am unfortunately forced to recommend the manuscript be rejected at this stage.
